# Efficient Detection of LLM-generated Texts with a Bayesian Surrogate Model

## Abstract

The detection of machine-generated text, especially from large language models (LLMs), is crucial in preventing serious social problems resulting from their misuse. Some methods train dedicated detectors on specific datasets but fall short in generalizing to unseen test data, while other zero-shot ones often yield suboptimal performance. Although the recent DetectGPT has shown promising detection performance, it suffers from significant inefficiency issues, as detecting a single candidate requires querying the source LLM with hundreds of its perturbations. This paper aims to bridge this gap. Concretely, we propose to incorporate a Bayesian surrogate model, which allows us to select typical samples based on Bayesian uncertainty and interpolate scores from typical samples to other samples, to improve query efficiency. Empirical results demonstrate that our method significantly outperforms existing approaches under a low query budget. Notably, our method achieves similar performance with up to 2 times fewer queries than DetectGPT and 3.7% higher AUROC at a query number of 5.

## 1 Introduction

Large language models (LLMs) (Brown et al., 2020; Chowdhery et al., 2022; OpenAI, 2022; Touvron et al., 2023) have the impressive ability to replicate human language patterns, producing text that appears coherent, well-written, and persuasive, although the generated text may contain factual errors and unsupported quotations. As LLMs are increasingly used to simplify writing and presentation tasks, some individuals regrettably have misused LLMs for nefarious purposes, such as creating convincing fake news articles or engaging in cheating, which can have significant social consequences. Mitigating these negative impacts has become a pressing issue for the community.

The LLM-generated texts are highly articulate, posing a significant challenge for humans in identifying them (Gehrmann et al., 2019b). Fortunately, it is shown that machine learning tools can be leveraged to recognize the watermarks underlying the texts. Some methods (e.g., OpenAI, 2023b) involve training supervised classifiers, which, yet, suffer from overfitting to the training data and ineffectiveness to generalize to new test data. *Zero-shot* LLM-generated text detection approaches bypass these issues by leveraging the source LLM to detect its samples (Solaiman et al., 2019; Gehrmann et al., 2019b; Ippolito et al., 2020). They usually proceed by inspecting the average per-token log probability of the candidate text, but the practical detection performance can be unsatisfactory.

DetectGPT (Mitchell et al., 2023) is a recent method that achieves improved zero-shot detection efficacy by exploring the probability curvature of LLMs. It generates multiple perturbations of the candidate text and scores them using the source LLM to define detection statistics. It can detect texts generated by GPT-2 (Radford et al., 2019) and GPT-NeoX-20B (Black et al., 2022). Yet, DetectGPT relies on hundreds of queries to the source LLM to estimate the local probability curvature surrounding *one* single candidate passage. This level of computational expense is impractical for handling large LMs like LLaMA (Touvron et al., 2023), ChatGPT (OpenAI, 2022), and GPT-4 (OpenAI, 2023a).

This paper aims to improve the query efficiency of probability curvature-based detectors. We highlight that the inefficiency issues of the current approach stem from the use of purely random perturbations for curvature estimation. Intuitively, due to the constraint on locality, the perturbed samples are highly correlated, sharing most words and having similar semantics. As a result, characterizing the local probability curvature can be essentially optimized as (*i*) identifying a set of typical samples and evaluating the source LLM on them and (*ii*) interpolating the results to other samples.

A surrogate model that maps samples to LLM probability is necessary for interpolation, and it would be beneficial if the model could also identify typical samples. Given these, we opt for the Gaussian process (GP) model due to its non-parametric flexibility, resistance to overfitting in low-data regimes, and ease of use in solving regression problems. More importantly, the Bayesian uncertainty provided by GP can effectively indicate sample typicality, as demonstrated in active learning (Gal et al., 2017). Technically, we perform sample selection and GP fitting sequentially. At each step, we select the sample that the current GP model is most uncertain about, score it using the source LLM, and update the GP accordingly. Early stops can be invoked adaptively. After fitting, we utilize the GP, rather than the source LLM, to score a set of randomly perturbed samples to compute detection statistics. This way, we create a zero-shot LLM-generated text detector with significantly improved query efficiency.

To showcase the effectiveness and efficiency of our method, we conduct comprehensive empirical studies on diverse datasets using GPT-2 (Radford et al., 2019) and LLaMA-65B (Touvron et al., 2023). The results show that our method outperforms DetectGPT with substantial margins under the low query budget—it can achieve similar performance with up to 2 times fewer queries than DetectGPT and achieve 3.7% higher AUROC at a query number of 5. This is one significant step toward *practical* use. We also show that our approach remains effective even when the logits are entirely invisible and conduct ablation studies to offer insights into the behavior of our method.

## 2 RELATED WORKS

**Large language models.** LLMs (Radford et al., 2019; Brown et al., 2020; Chowdhery et al., 2022; Zhang et al., 2022; OpenAI, 2022) have revolutionized the field of natural language processing by offering several advantages over previous pre-trained models (Devlin et al., 2018; Liu et al., 2019; Lan et al., 2019), including a better characterization of complex patterns and dependencies in the text, and the appealing in-context learning ability for solving downstream tasks with minimal examples. Representative models such as GPT-3 (Brown et al., 2020), PaLM (Chowdhery et al., 2022), and ChatGPT (OpenAI, 2022) have showcased their remarkable ability to generate text with high coherence, fluency, and semantic relevance. They can even effectively address complex inquiries related to science, mathematics, history, current events, and social trends. Therefore, it is increasingly important to effectively regulate the use of LLMs to prevent significant social issues.

**LLM-generated text detection.** Previous methods can be broadly categorized into two groups. The first group of methods performs detection in a zero-shot manner (Solaiman et al., 2019; Gehrmann et al., 2019a; Mitchell et al., 2023), but they require access to the source model that generates the texts to derive quantities like output logits or losses for detection. For instance, Solaiman et al. (2019) suggest that a higher log probability for each token indicates that the text will likely be machine-generated. When the output logits/losses of the source model are unavailable, these methods rely on a proxy model for detection. However, there is often a substantial gap between the proxy and source models from which the text is generated. Another group of methods trains DNN-based classifiers on collected human-written and machine-generated texts for detection (Guo et al., 2023; Uchendu et al., 2020; OpenAI, 2023b). However, such detectors are data-hungry and may exhibit poor generalization ability when facing domain shift (Bakhtin et al., 2019; Uchendu et al., 2020). Furthermore, training DNN-based classifiers is susceptible to backdoor attacks (Qi et al., 2021) and adversarial attacks (He et al., 2023). Besides, both He et al. (2023) and Li et al. (2023) develop benchmarks for evaluating existing detection methods and call for more robust detection methods.

## 3 METHODOLOGY

This section first reviews DetectGPT (Mitchell et al., 2023) and highlights its query inefficiency to justify the need for a surrogate model. We then emphasize the importance of being Bayesian and provide a comprehensive discussion on the specification of the surrogate model. We also discuss the strategy for selecting typical samples with Bayesian uncertainty. The method overview is in Fig. 1.

### 3.1 PRELIMINARY

We consider the zero-shot LLM-generated text detection problem, which essentially involves binary classification to judge whether a text passage originates from a language model or not. Zero-shot detection implies we do not require a dataset composed of human-written and LLM-generated texts to train the detector. Instead, following common practice (Solaiman et al., 2019; Gehrmann et al.,

| ⬟ Candidate text passage $x$ | ⬤ Typical perturbations from $q(\cdot|x)$ | —— $\log p_\theta$ of the source LLM | - - - Surrogate model | ▨ Uncertainty |

Figure 1: Method overview. Following DetectGPT (Mitchell et al., 2023), we explore the local structure of the probability curvature of the LLM $p_\theta$ to determine whether a text passage $x$ originates from it. However, instead of using the source LLM to score numerous random perturbations, we leverage the high redundancy among these perturbations to enhance query efficiency. We select a limited number of typical samples for scoring and interpolate their scores to other samples. To achieve reasonable selection and interpolation, we employ a Gaussian process as the surrogate model, which, as shown, enjoys non-parametric flexibility and delivers calibrated uncertainty in the presence of a suitable kernel. The figure above also demonstrates the sequential selection of typical samples—at each step, the sample that the surrogate model is most uncertain about is chosen. After fitting, we use the surrogate model as a substitute for $\log p_\theta$ to calculate the detection measure $\ell(x, p_\theta, q)$ in Eq. (2).

2019b; Ippolito et al., 2020; Mitchell et al., 2023), we assume access to the source LLM to score the inputs, based on which the detection statistics are constructed.

DetectGPT (Mitchell et al., 2023) is a representative work in this line. It utilizes the following measure to determine if a text passage $x$ is generated from a large language model $p_\theta$:

$$\log p_\theta(x) - \mathbb{E}_{\tilde{x} \sim q(\cdot|x)} \log p_\theta(\tilde{x}), \tag{1}$$

where $q(\cdot|x)$ is a perturbation distribution supported on the semantic neighborhood of the candidate text $x$. For example, $q(\cdot|x)$ can be defined with manual rephrasings of $x$ while maintaining semantic similarity. DetectGPT, in practice, instantiates $q(\cdot|x)$ with off-the-shelf pre-trained mask-filling models like T5 (Raffel et al., 2020) to avoid humans in the loop.

For tractablility, DetectGPT approximates Eq. (1) with Monte Carlo samples $\{x_i\}_{i=1}^N$ from $q(\cdot|x)$:

$$\log p_\theta(x) - \frac{1}{N} \sum_{i=1}^N \log p_\theta(x_i) =: \ell(x, p_\theta, q). \tag{2}$$

Based on the hypothesis that LLM-generated texts should locate in the local maxima of the log probability of the source LLM, it is expected that $\ell(x, p_\theta, q)$ is large for LLM-generated texts but small for human-written ones, and thus a detector emerges. Despite good performance, DetectGPT is costly because detecting one single candidate text hinges on $N + 1$ (usually, $N \geq 100$) queries to the source model $p_\theta$, which can lead to prohibitive overhead when applied to commercial LLMs.

## 3.2 IMPROVE QUERY EFFICIENCY WITH A SURROGATE MODEL

Intuitively, we indeed require numerous perturbations to reliably estimate the structure of the local probability curvature around the candidate text $x$ due to the high dimensionality of texts. However, given the mask-filling nature of the perturbation model and the requirement for semantic locality, the samples to be evaluated, referred to as $\mathbf{X} = \{x_i\}_{i=0}^N$,[1] share a substantial amount of content and semantics. Such significant redundancy and high correlation motivate us to select only a small set of typical samples for scoring by the source LLM and then reasonably interpolate the scores to other samples (see Fig. 1). This way, we obtain the detection measure in a query-efficient manner.

A surrogate model that maps samples to LLM probability is required for interpolation, which should also help identify typical samples if possible. The learning of the model follows a standard regression setup. Let $\mathbf{X}_t = \{x_{t_i}\}_{i=0}^T \subset \mathbf{X}$,[2] denote a subset of typical samples selected via some tactics and $y_t = \{\log p_\theta(x_{t_i})\}_{i=0}^T$ the corresponding log-probabilities yielded by the source LLM. The surrogate model $f : \mathcal{X} \to \mathbb{R}$ is expected to fit the mapping from $\mathbf{X}_t$ to $y_t$, while being able to generalize reasonably to score the other samples in place of the source LLM.

---

[1] If not misleading, $x_0$ denotes the original candidate text $x$.

[2] We constrain $t_0 = 0$ to include the original text $x_0$ in $\mathbf{X}_t$.

### 3.3 The Bayesian Surrogate Model

Before discussing how to select the typical samples, it is necessary to clarify the specifications of the surrogate model. In our approach, we fit a dedicated surrogate model for each piece of text $\boldsymbol{x}$, and it approximates the source LLM only in the local region around $\boldsymbol{x}$. This allows us to avoid the frustrating difficulty of approximating the entire probability distribution represented by the source LLM, and work with lightweight surrogate models as well as good query efficiency.

The surrogate model $f$ is expected to be trained in the low-data regime while being expressive enough to handle non-trivial local curvature and not prone to overfitting. Additionally, the model should inherently incorporate mechanisms for typical sample selection. Given these requirements, we chose to use a Gaussian process (GP) model as the surrogate model due to its non-parametric flexibility, resistance to overfitting, and capability to quantify uncertainty (Williams & Rasmussen, 1995). Parametric models such as neural networks cannot meet all of these requirements simultaneously.

Concretely, consider a GP prior in function space, $f(\boldsymbol{x}) \sim \mathcal{GP}(0, k(\boldsymbol{x}, \boldsymbol{x}'))$, where the mean function is set to zero following common practice and $k(\boldsymbol{x}, \boldsymbol{x}')$ refers to the kernel function. Consider a Gaussian likelihood with noise variance $\sigma^2$ for this problem, i.e., $y(\boldsymbol{x})|f(\boldsymbol{x}) \sim \mathcal{N}(y(\boldsymbol{x}); f(\boldsymbol{x}), \sigma^2)$.

**Posterior predictive**. It is straightforward to write down the posterior distribution of the function values $\boldsymbol{f}_{\mathbf{X}^*}$ for unseen samples $\mathbf{X}^* = \{\boldsymbol{x}_i^*\}_{i=1}^M$, detailed below

$$p(\boldsymbol{f}_{\mathbf{X}^*}|\mathbf{X}_t, \boldsymbol{y}_t, \mathbf{X}^*) = \mathcal{N}(\bar{\boldsymbol{f}}^*, \boldsymbol{\Sigma}^*) \tag{3}$$

where

$$\begin{aligned}
\bar{\boldsymbol{f}}^* &:= \boldsymbol{k}_{\mathbf{X}^*, \mathbf{X}_t}[\boldsymbol{k}_{\mathbf{X}_t, \mathbf{X}_t} + \sigma^2 \mathbf{I}]^{-1} \boldsymbol{y}_t \\
\boldsymbol{\Sigma}^* &:= \boldsymbol{k}_{\mathbf{X}^*, \mathbf{X}^*} - \boldsymbol{k}_{\mathbf{X}^*, \mathbf{X}_t}[\boldsymbol{k}_{\mathbf{X}_t, \mathbf{X}_t} + \sigma^2 \mathbf{I}]^{-1} \boldsymbol{k}_{\mathbf{X}_t, \mathbf{X}^*}.
\end{aligned} \tag{4}$$

$\boldsymbol{k}_{\mathbf{X}^*, \mathbf{X}_t} \in \mathbb{R}^{M \times (T+1)}, \boldsymbol{k}_{\mathbf{X}_t, \mathbf{X}_t} \in \mathbb{R}^{(T+1) \times (T+1)}, \boldsymbol{k}_{\mathbf{X}^*, \mathbf{X}^*} \in \mathbb{R}^{M \times M}$ are evaluations of the kernel $k$. With this, we can analytically interpolate scores from the typical samples to new test samples.

**Text kernel.** It should be noted that the GP model described above is designed to operate within the domain of natural language, meaning traditional kernels like RBF and polynomial kernels are not suitable. To address this challenge, we draw inspiration from BertScore (Zhang et al., 2019), which has demonstrated a good ability to capture similarities between text passages. We make a straightforward modification to BertScore, resulting in the following kernel:

$$k(\boldsymbol{x}, \boldsymbol{x}') := \alpha \cdot \text{BertScore}(\boldsymbol{x}, \boldsymbol{x}') + \beta \tag{5}$$

where $\alpha \in \mathbb{R}^+$ and $\beta \in \mathbb{R}$ are two hyperparameters to boost flexibility. Other symmetric positive semi-definite kernels defined on texts are also applicable here.

**Hyperparameter tuning.** To make the hyperparameters $\alpha, \beta$, and $\sigma^2$ suitable for the data at hand, we optimize them to maximize the log marginal likelihood of the targets $\boldsymbol{y}_t$, a canonical objective for hyperparameter tuning for Bayesian methods:

$$\log p(\boldsymbol{y}_t|\mathbf{X}_t, \alpha, \beta, \sigma^2) \propto -[\boldsymbol{y}_t^\top (\boldsymbol{k}_{\mathbf{X}_t, \mathbf{X}_t} + \sigma^2 \mathbf{I})^{-1} \boldsymbol{y}_t + \log |\boldsymbol{k}_{\mathbf{X}_t, \mathbf{X}_t} + \sigma^2 \mathbf{I}|], \tag{6}$$

where $\alpha$ and $\beta$ exist in the computation of $\boldsymbol{k}_{\mathbf{X}_t, \mathbf{X}_t}$. We can make use of AutoDiff libraries to perform gradient-based optimization of the hyperparameters directly. Since the number of samples in $\mathbf{X}_t$ is typically less than 100, the computational resources required for calculating matrix inversion and log-determinant are negligible.

### 3.4 Sequential Selection of Typical Samples

As discussed above, once we obtain the set of typical samples, we can effortlessly score new samples using the GP model. We next describe how to use Bayesian uncertainty to identify the typical samples.

In our case, the typicality of a text sample depends on the surrogate model. If the surrogate model can accurately predict the score for the sample, it should not be considered typical, and vice versa. However, relying on the ground-truth score to measure typicality is query-intensive, and hence impractical.

Fortunately, in Bayesian methods, there is often a correlation between the model's prediction accuracy and uncertainty, with higher uncertainty implying lower accuracy (Lakshminarayanan et al., 2017;

---

**Algorithm 1** Efficient detection of LLM-generated texts with a Bayesian surrogate model.

---

1: **Input:** Text passage $\boldsymbol{x}$, LLM $p_{\boldsymbol{\theta}}$, perturbation model $q(\cdot|\boldsymbol{x})$, kernel $k(x, x')$, hyperparameters $\alpha, \beta, \sigma^2$, sample sizes $N, T, S$, detection threshold $\delta$.
2: **Output:** True/false indicating whether the text passage $\boldsymbol{x}$ comes from the LLM $p_{\boldsymbol{\theta}}$ or not.
3: Perform rephrasing with $q(\cdot|\boldsymbol{x})$, obtaining perturbations $\mathbf{X} = \{\boldsymbol{x}_i\}_{i=0}^N$;
4: Randomly initialize the typical set $\mathbf{X}_t$ and the set $\mathbf{X}^*$ for selection;
5: $\boldsymbol{y}_t \leftarrow \log p_{\boldsymbol{\theta}}(\mathbf{X}_t)$;
6: **while** $|\mathbf{X}_t| < T$ *or* other early stop criteria have not been satisfied **do**
7:     Optimize the hyperparameters $\alpha, \beta$, and $\sigma^2$ according to Eq. (6) given $\mathbf{X}_t$ and $\boldsymbol{y}_t$;
8:     Estimate the predictive covariance $\boldsymbol{\Sigma}^*$ for $\mathbf{X}^*$ detailed in Eq. (4);
9:     Identify the sample in $\mathbf{X}^*$ with the largest uncertainty (i.e., the diagonal element of $\boldsymbol{\Sigma}^*$);
10:     Score this sample with the LLM;
11:     Append the sample and the target to $\mathbf{X}_t$ and $\boldsymbol{y}_t$ respectively;
12: Approximately estimate the detection measure $\ell(\boldsymbol{x}, p_{\boldsymbol{\theta}}, q)$ with the resulting GP model;
13: **Return** True *if* $\ell(\boldsymbol{x}, p_{\boldsymbol{\theta}}, q) > \delta$ *else* False;

---

Maddox et al., 2019; Deng & Zhu, 2023). This allows us to leverage the Bayesian uncertainty of the employed GP model to select typical samples sequentially. We should choose samples with high predictive uncertainty, i.e., samples the model is likely to predict inaccurately. Interestingly, such an uncertainty-based selection strategy has similarities with those employed in existing active learning approaches (Gal et al., 2017; Mohamadi & Amindavar, 2020), highlighting the soundness of our approach. Our approach also resembles a Bayesian optimization program that finds points maximizing an acquisition function (Snoek et al., 2012). In our case, the acquisition contains only the uncertainty term.

Specifically, we perform data selection and model fitting alternately, with the following details.

**Initlization.** We initialize $\mathbf{X}_t$ with a random subset of $\mathbf{X}$, i.e., $\mathbf{X}_t = \{\boldsymbol{x}_{t_i}\}_{i=0}^S, 1 \leq S < T$. Unless otherwise specified, we set $S = 2$, where the first sample refers to the original candidate text and the second a random perturbation of it. We use $\boldsymbol{y}_t$ to denote the corresponding ground-truth scores.

**Model fitting.** Optimize the hyperparameters of the GP model on data $(\mathbf{X}_t, \boldsymbol{y}_t)$ to maximize the log marginal likelihood defined in Eq. (6).

**Data selection.** Denote by $\mathbf{X}^*$ the samples for selection. It can be the complement of $\mathbf{X}_t$ in $\mathbf{X}$ or other random perturbations around the candidate $\boldsymbol{x}$. Compute the covariance matrix $\boldsymbol{\Sigma}^*$ defined in Eq. (4), whose diagonal elements correspond to the predictive uncertainty of samples in $\mathbf{X}^*$. Augment the sample with the largest uncertainty to $\mathbf{X}_t$, and append its score yielded by the source LLM to $\boldsymbol{y}_t$.

**Adaptive exit.** If the size of $\mathbf{X}_t$ equals $T$ or a specific stop criterion is satisfied, e.g., the largest uncertainty is lower than a threshold, the program exits from the model fitting and data selection loop.

**Estimation of detection measure.** We use the resulting GP model to compute the approximate log probability for all samples in $\mathbf{X}$, and hence get an estimation of the detection measure $\ell(\boldsymbol{x}, p_{\boldsymbol{\theta}}, q)$.

We depict the overall algorithmic procedure in Algorithm 1. We clarify our method also applies to situations where only a proxy of the source LLM is available, as demonstrated in Section 4.3. Despite decreasing the number of queries to the source LLM, our method needs to estimate the BertScore in the local environment frequently.

## 4 EXPERIMENTS

We conduct extensive experiments to evaluate the efficiency and efficacy of our method for zero-shot LLM-generated text detection. We mainly compare our method to DetectGPT (Mitchell et al., 2023) because (*i*) both works adopt the same detection measure, and (*ii*) DetectGPT has proven to defeat prior zero-shot and supervised methods consistently. We are primarily concerned with detecting under a low query budget and admit that our method would perform similarly to DetectGPT if queries to the source model can be numerous. We also consider a black-box variant of the task where only a proxy of the source LLM is available for detection. We further qualitatively analyze the behavioral difference between our method and DetectGPT and showcase the limitations of our method.

**Datasets.** Following DetectGPT (Mitchell et al., 2023), we primarily experiment on three datasets, covering news articles sourced from the XSum dataset (Narayan et al., 2018), which represents the

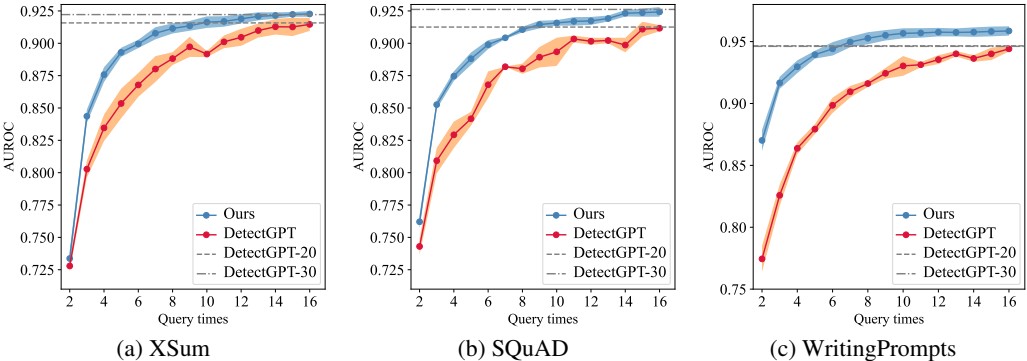

(a) XSum        (b) SQuAD        (c) WritingPrompts

Figure 2: The AUROC for detecting samples generated by GPT-2 varies depending on the number of queries made to the source GPT-2. We present the results on three representative datasets.

problem of identifying fake news, paragraphs from Wikipedia drawn from the SQuAD contexts (Rajpurkar et al., 2016), which simulates the detection of machine-generated academic essays, and prompted stories from the Reddit WritingPrompts dataset (Fan et al., 2018), which indicates the recognition of LLM-created creative writing submissions. These datasets are representative of a variety of common domains and use cases for LLM. We let the LLMs expand the first 30 tokens of the real text to construct generations. Refer to Mitchell et al. (2023) for more details.

**Evaluation Metric.** The detection of LLM-generated texts is actually a binary classification problem. Thereby, we use the area under the receiver operating characteristic curve (AUROC) as the key metric to evaluate the performance of the corresponding detectors (i.e., classifiers).

**The LLMs of concern.** We focus on two widely used open-source LLMs: GPT-2 (Radford et al., 2019) and LLaMA-65B (Touvron et al., 2023). GPT-2 is an LLM that leverages the strengths of the GPT architecture. On the other hand, LLaMA-65B has gained more attention recently for its ability to build customized chatbots. It is the largest variant in the LLaMA family.

**Hyperparameters.** Most hyperparameters regarding the perturbation model, i.e., $q(\cdot|\boldsymbol{x})$, follow DetectGPT (Mitchell et al., 2023). We use T5-large when detecting samples from GPT-2 and T5-3B when detecting samples from LLaMA-65B. Unless otherwise specified, we set the sample size $N$ for estimating the detection measure to 200 and $S$ to 2. We tune the hyperparameters associated with the GP model with an Adam optimizer (Kingma & Ba, 2014) using a learning rate of 0.01 (cosine decay is deployed) for 50 iterations.

### 4.1 DETECTION OF GPT-2

We first compare our method to DetectGPT in detecting contents generated by GPT-2.

Specifically, we evaluate the detection performance of DetectGPT and our method with the query budget continually increasing. Letting $Q$ denote the query budget, DetectGPT uses $Q-1$ random perturbations to estimate the detection measure $\ell(\boldsymbol{x}, p_{\boldsymbol{\theta}}, q)$. In contrast, our method uses $Q-1$ typical samples for fitting the surrogate model and still uses a large number of random perturbations (as stated, 200) to estimate $\ell$, which is arguably more reliable.

In Fig. 2, we present a comparison of detection AUROC on the datasets mentioned earlier. To reduce the influence of randomness, we report average results and variances over three random runs. For a comprehensive comparison, we also draw the performance of DetectGPT using 20 and 30 queries in the figure. As shown, our method outperforms DetectGPT significantly, achieving faster performance gains, particularly under a lower query budget. Notably, our method using only 10 queries can outperform DetectGPT using 20 queries in all three cases.

Interestingly, our method can already surpass DetectGPT in the 2-query case, especially on the WritingPrompts dataset. In that case, our method fits a GP model with only the original text passage as well as a random perturbation. This result confirms that the GP-based surrogate model is highly data-efficient and excels at interpolating the scores of typical samples to unseen data. Furthermore, DetectGPT's performance gain with increasing query times is slow on the WritingPrompts dataset, as its detection AUROCs using 16, 20, and 30 queries are similar. In contrast, our method does not face

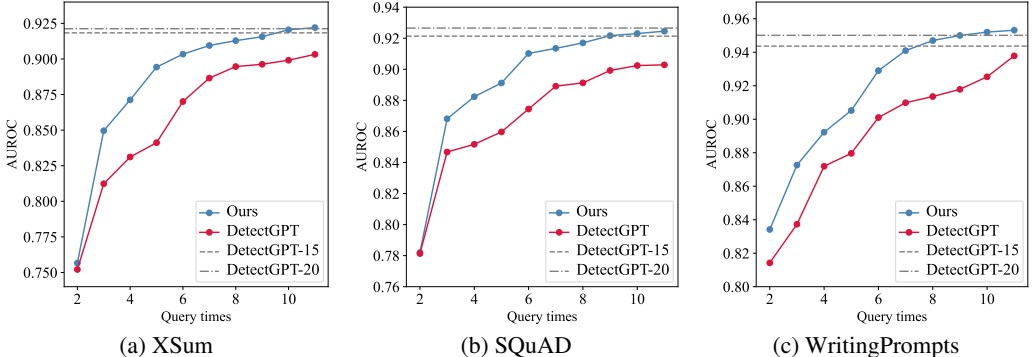

Figure 3: The AUROC for detecting samples generated by GPT-2 varies depending on the number of queries made to the source GPT-2. We use T5-3B as the perturbation model here.

Table 1: Comparison on TPR at various FPRs for detecting samples generated by GPT-2 with query budget 15.

| Method | Xsum | | Squad | | Writing | |
|---|---|---|---|---|---|---|
| | FPR@0.01 | FPR@0.05 | FPR@0.01 | FPR@0.05 | FPR@0.01 | FPR@0.05 |
| DetectGPT | 0.278 | 0.514 | 0.230 | 0.487 | 0.420 | 0.678 |
| Our Method | 0.314 | 0.574 | 0.267 | 0.613 | 0.502 | 0.784 |

this issue. Given that the variances are minor compared to the mean values, we omit repeated random runs in the following analysis.

**Impact of the perturbation model.** The perturbation model used in the above studies is the T5-large model. We then question whether replacing it with a more powerful model results in higher detection performance. For an answer, we introduce the T5-3B model as the perturbation model and conduct a similar set of experiments, with the results displayed in Fig. 3.

As shown, the detection performance of DetectGPT and our method is substantially improved compared to the results in Fig. 2, indicating the higher ability of T5-3B to perturb in the semantic space. Still, our method consistently surpasses DetectGPT and achieves similar performance with up to $2\times$ fewer queries than DetectGPT. In particular, the average AUROCs of our method at query times of 5 and 10 are $0.897$ and $0.932$, respectively, while those for DetectGPT are $0.860$ and $0.909$.

We also inspect the True Positive Rate (TPR) at various False Positive Rates (FPRs) following (Sadasivan et al., 2023). As shown in Table 1, at low FPRs like $0.01$ or $0.05$, our approach can significantly outperform DetectGPT. Besides, we plot the corresponding ROC curves in Appendix. Our detector can achieve 83.6% TPR on XSum, 87.0% on SQuAD, and 92.8% TPR on WritingPrompts at 15% FPR when the query budget is $15$.

**High query budget.** To demonstrate that our method remains effective with more query times, we test our method on 50 queries for case study, and it achieved a final AUROC of 98.0% on WritingPrompts, the same as DetectGPT using about 150 queries. This result verifies the much higher efficiency of our method than DetectGPT while being able to achieve state-of-the-art performance.

## 4.2    DETECTION OF LLAMA-65B

As GPT-2 is limited in model size and capacity, we extend our evaluation to the LLaMA family (Touvron et al., 2023) and focus on the largest variant, LLaMA-65B, to thoroughly investigate the practical application value of our method. LLaMA-65B is trained on a diverse range of web text and conversational data and has gained recent attention. The texts generated by LLaMA-65B are of exceptional quality and coherence, closely resembling texts written by humans.

Given the previous studies, we use T5-3B as the perturbation model in this case. Due to the non-trivial resource consumption of deploying LLaMA-65B, we consider the range of query times only up to 11. Nevertheless, we still include DetectGPT under $15$ and $20$ query budgets to emphasize the query-saving effects of our method. We display the comparison between DetectGPT and our method in Fig. 4, where the three datasets and various query budgets are also considered.

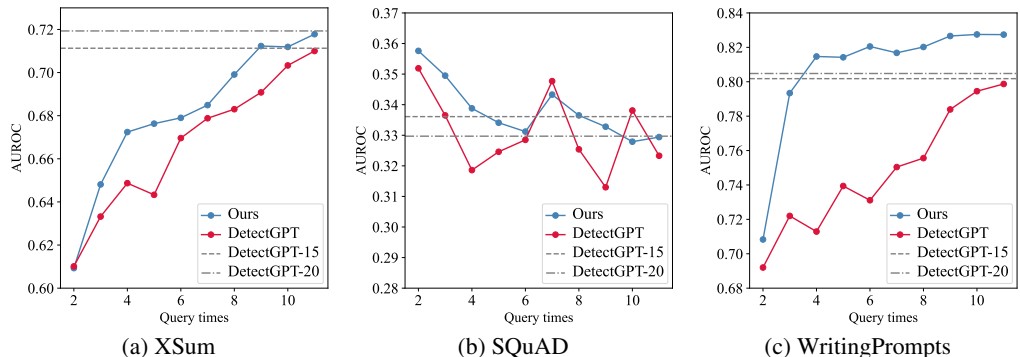

Figure 4: The AUROC for detecting samples generated by LLaMA-65B varies depending on the number of queries made to the source LLaMA-65B. We use T5-3B as the perturbation model.

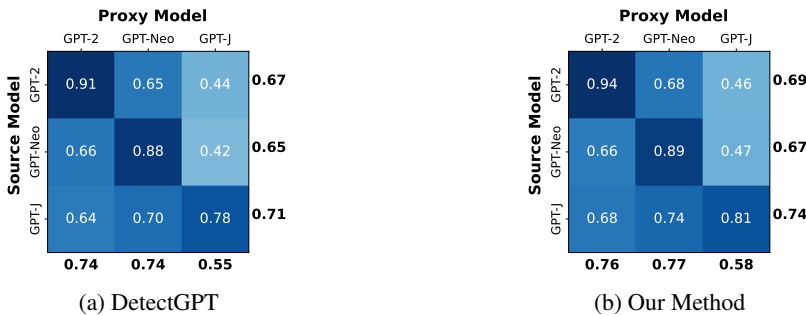

Figure 5: Cross evaluation of using various source models (i.e., those generating the texts) and proxy models (i.e., those scoring the texts for detection) in detection. We select models from {GPT-J, GPT-Neo-2.7, GPT-2}. We report the average AUROC over the three datasets. We offer the row/column mean. The query budget is 15.

As shown, our method continues to outperform DetectGPT on XSum and WritingPrompts. Notably, in WritingPrompts, the performance gain of our method is even more substantial than in previous experiments. With just 4 queries, our method exceeds DetectGPT's performance with 20 queries by a significant margin. We also find DetectGPT has difficulties realizing better detection under a higher query budget in this case, implying the inherent inefficiency of random perturbation-based detection.

On SQuAD, both our method and DetectGPT give a detection AUROC below 50%. This suggests that the probability curvature hypothesis postulated in Mitchell et al. (2023) does not apply to the texts generated by LLaMA-65B based on the initial Wikipedia paragraphs sourced from SQuAD contexts. The possible reasons include (1) the generations do not locate around the local maxima of the model likelihood of LLaMA-65B; (2) the real texts in this dataset instead have a higher model likelihood. We visualize some real texts and generated ones by LLaMA-65B in Appendix and found it hard to draw an intuitive conclusion. We leave an in-depth investigation as future work.

## 4.3 CROSS EVALUATION

The above studies assume a white-box setting, where the source LLM is utilized to detect its generations. Yet, in practice, we may not know which model the candidate passage was generated from. A remediation is to introduce a proxy model for an approximate estimate of the log probability of source LLM (Mitchell et al., 2023). This section examines the impact of doing this on the final detection performance. To reduce costs, we consider using GPT-J (Wang & Komatsuzaki, 2021), GPT-Neo-2.7 (Black et al., 2021), and GPT-2 as the source and proxy models and evaluate the detection AUROC across all 9 source-proxy combinations. We present the average performance across 200 samples from XSum, SQuAD, and WritingPrompts in Fig. 5.

As demonstrated, using the same model for both generating and scoring yields the highest detection performance, but employing a scoring model different from the source model can still be advantageous. The column mean represents the quality of a scoring model, and our results suggest that GPT-Neo-2.7 is more effective in accounting for scoring. Furthermore, we observe significant improvements (up to 3% AUROC) in our method's results over DetectGPT, indicating the generalizability of our approach.

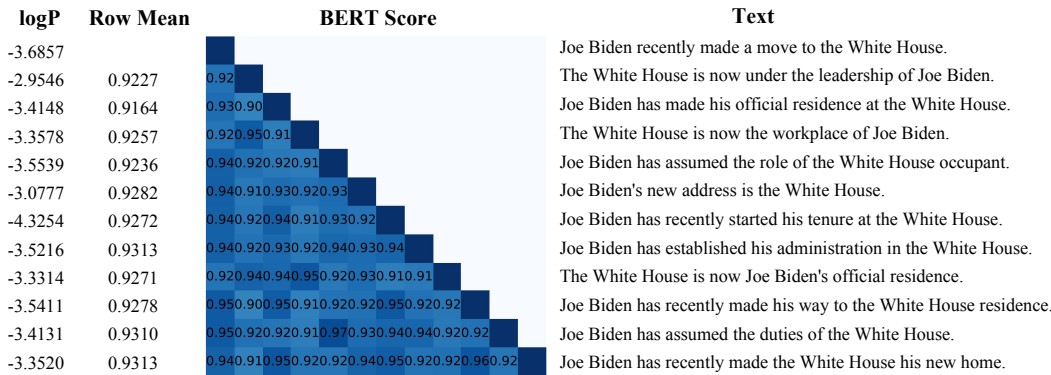

| logP | Row Mean | BERT Score | Text |
|---|---|---|---|
| -3.6857 | | | Joe Biden recently made a move to the White House. |
| -2.9546 | 0.9227 | | The White House is now under the leadership of Joe Biden. |
| -3.4148 | 0.9164 | | Joe Biden has made his official residence at the White House. |
| -3.3578 | 0.9257 | | The White House is now the workplace of Joe Biden. |
| -3.5539 | 0.9236 | | Joe Biden has assumed the role of the White House occupant. |
| -3.0777 | 0.9282 | | Joe Biden's new address is the White House. |
| -4.3254 | 0.9272 | | Joe Biden has recently started his tenure at the White House. |
| -3.5216 | 0.9313 | | Joe Biden has established his administration in the White House. |
| -3.3314 | 0.9271 | | The White House is now Joe Biden's official residence. |
| -3.5411 | 0.9278 | | Joe Biden has recently made his way to the White House residence. |
| -3.4131 | 0.9310 | | Joe Biden has assumed the duties of the White House. |
| -3.3520 | 0.9313 | | Joe Biden has recently made the White House his new home. |

Figure 6: The visualization of the candidate text passage and the first 11 typical perturbations of it identified by our method, ordered from top to bottom. The BertScore among them and the row mean (estimated without the diagonal elements) are reported. The log probabilities are given by GPT-2.

## 4.4 MORE STUDIES

To better understand our method, we lay out the following additional studies.

**How does our method behave differently from DetectGPT?** We are interested in how our method achieves better detection performance than DetectGPT. To chase an intuitive answer, we collect some human-written texts from the considered three datasets as well as the generated texts from GPT-2, and compute the detection measure $\ell(x, p_\theta, q)$ estimated by DetectGPT and our method under a query budget of 15 and the T5-large perturbation model. We list the results in the Appendix due to space constraints. We find that (1) Our method's estimation of $\ell$ is usually higher than DetectGPT. Recalling the expression of $\ell$, we conclude our method can usually select samples with substantially lower $\log p_\theta$ than random samples. (2) Our method can occasionally produce too high or too low $\ell$ for texts written by humans. This could be attributed to the fact that using a limited number of typical samples may fail to reliably capture the local curvature when it is ill-posed or complex.

**The visualization of typical samples.** As depicted in Fig. 1, typical samples have a direct physical meaning in toy cases. However, it is unclear whether this property can be maintained in the high-dimensional space in which texts exist. To investigate this, we conducted a simple study based on the text passage, "Joe Biden recently made a move to the White House." Specifically, we perturb it with the rewriting function of ChatGPT (OpenAI, 2022) for 50 times, and simulate the sequential procedure of typical sample selection. We present the original text and the first 11 typical samples in Fig. 6. We also display the BertScore among them as well as the log probabilities $\log p_\theta$ of GPT-2 for them. As shown, the row mean exhibits a clear increasing trend as the index of the typical sample increases, indicating that the later selected samples are becoming more similar to the earlier ones. In other words, the uniqueness or typicality of the selected samples decreases over the course of the selection process. This phenomenon is consistent with our expectations for the selected samples and confirms the effectiveness of our uncertainty-based selection method.

## 5 CONCLUSION AND SOCIAL IMPACT

This paper tackles the issue that the existing probability curvature-based method for detecting LLM-generated text relies on a vast number of queries to the source LLM when detecting a candidate text passage. We introduce a Bayesian surrogate model to identify typical samples and then interpolate their scores to others. We use a Gaussian process regressor to instantiate the surrogate model and perform an online selection of typical samples based on Bayesian uncertainty. Extensive empirical studies on various datasets, using GPT-2 and LLaMA-65B, validate our method's superior effectiveness and efficiency over DetectGPT.

A limitation is that our method is not compatible with parallel computing due to the sequential nature of sample selection. Besides, with the goal of enhancing DetectGPT's query efficiency, we have not evaluated its reliability against paraphrasing attacks. Regarding the social impact, we emphasize that detecting LLM-generated text is crucial in preventing serious social problems that may arise from the misuse of LLMs. E.g., LLM-generated text could be used to spread false information, manipulate public opinion, or even incite violence. Effective detection approaches are crucial in addressing these.

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

## A   ROC Curves

To better demonstrate the effectiveness of our approach, we analyzed the True Positive Rate (TPR) at various False Positive Rates (FPRs) following (Sadasivan et al., 2023). Fig 7 shows the ROC curves for detecting samples generated by GPT-2 when the query budget is 15 with T5-3B as the perturbation model. Our detector can achieve 83.6% TPR on XSum, 87.0% on SQuAD, and 92.8% TPR on WritingPrompts at 15% FPR. We also have a comparison to DetectGPT of TPR at low FPRs in Table 1. The results clearly demonstrate that our method consistently outperforms DetectGPT across various scenarios.

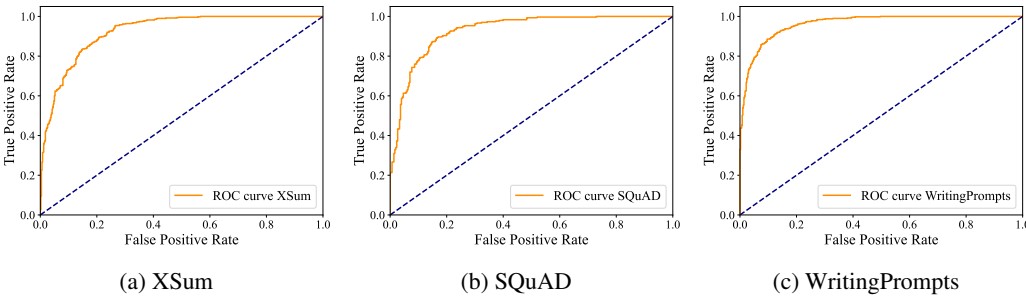

| (a) XSum | (b) SQuAD | (c) WritingPrompts |

Figure 7: The ROC curves for detecting samples generated by GPT-2 when query budget is 15. We present the results on three representative datasets. We use T5-3B as the perturbation model.

## B   Comparison between Real Texts and Generated Texts from LLaMA-65B on SQuAD

We list some randomly selected texts in Fig. 8, where the first 30 tokens of the real texts (highlighted in red) are used to prompt generation.

## C   The Behavioral Difference between DetectGPT and Our Method

We collect some human-written texts and generated texts from GPT-2, meanwhile computing the detection measure $\ell(\boldsymbol{x}, p_{\boldsymbol{\theta}}, q)$ estimated by DetectGPT and our method under a query budget of 15 and the T5-large perturbation model. We present the results in Table 2.

| Human-written passage | Generations from LLaMA-65B |
|---|---|
| An adolescent's environment plays a huge role in their identity development. While most adolescent studies are conducted on white, middle class children, studies show that the more privileged upbringing people have, the more successfully they develop their identity. The forming of an adolescent's identity is a crucial time in their life... | An adolescent's environment plays a huge role in their identity development. While most adolescent studies are conducted on white, middle class, suburban youth; a large number of immigrant, working class youth live in urban environments similar to that of my participants. Through participant observation, interviews, and textual analysis... |
| Compared to other team sports, the National Football League preseason is very structured. Every NFL team plays exactly four pre-season exhibition games a year, two at home and two away, with the exception of two teams each year who play a fifth game, the Pro Football Hall of Fame Game.These exhibition games, most of which are held in the month of August... | Compared to other team sports, the National Football League preseason is very structured. Every NFL team plays exactly four pre-season exhibition games \u2013 two games at home which are played at their own stadium, and two games away which are played at NFL stadiums at distant locations. In fact, NFL preseason games are played abroad with teams... |
| The Treaty of Guadalupe Hidalgo, signed on February 2, 1848, by American diplomat Nicholas Trist and Mexican plenipotentiary representatives Luis G. Cuevas, Bernardo Couto, and Miguel Atristain, ended the war, gave the U.S. undisputed control of Texas, and established the U.S.\u2013Mexican border of the Rio Grande. As news of peace negotiations reached ... | The Treaty of Guadalupe Hidalgo, signed on February 2, 1848, by American diplomat Nicholas Trist and Mexican diplomat Luis Gonzaga Cuevas, acknowledged Mexican relinquishment of about half of the extensive territory of the \"Mexican Cession\" Posts Tagged \u2018Ferry Corsten\u2019\nLive: Vondelpark Festival @ Vondelpark, Amsterdam... |
| Sound could be stored in either analog or digital format and in a variety of surround sound formats; NTSC discs could carry two analog audio tracks, plus two uncompressed PCM digital audio tracks, which were (EFM, CIRC, 16-bit and 44.056 kHz sample rate). PAL discs could carry one pair of audio tracks, either analog or digital and the digital tracks on ... | Sound could be stored in either analog or digital format and in a variety of surround sound formats; NTSC discs could carry two analog surround channels while PAL discs could carry up to four. On each format, these were identified as either \u201cM&E\u201d (mono or stereo full band audio with or without the music) or \u201cP&E\u201d (dial norm full... |
| From his diagrams of a small number of particles interacting in spacetime, Feynman could then model all of physics in terms of the spins of those particles and the range of coupling of the fundamental forces. Feynman attempted an explanation of the strong interactions governing nucleons scattering called the parton model. The parton model emerged as a ... | From his diagrams of a small number of particles interacting in spacetime, Feynman could then model all of physics in terms of the interactions of virtual particles (actually, in the path-integral version of quantum mechanics this leads to wave-like objects, to avoid paradoxes between quantum mechanics and general relativity). The interaction of matter then becomes nothing more than... |

Figure 8: Real texts in SQuAD dataset vs. generated texts from LLaMA-65B.

Table 2: Comparison of the behavioral difference between DetectGPT and our method. The numbers reported refer to the measure $\ell(\boldsymbol{x}, p_{\boldsymbol{\theta}}, q)$ estimated under a query budget of 15 and the T5-large perturbation model. Note that each pair of human-written and LLM-generated texts has the same starting tokens. The exhibited samples are randomly selected.

| Text | DetectGPT | Ours |
|---|---|---|
| (Human-written)... Others, however, thought that "the most striking aspect of the series was the genuine talent it revealed". It was also described as a "sadistic musical bake-off", and "a romp in humiliation". Other aspects of the show have attracted criticisms. The product placement in the show in particular was noted... | 0.0631 | 0.0857 |
| (LLM-generated) ... it's frustratingly mediocre." While the reviewer on the Los Angeles Times site called American Idol "hilarious," adding "not much there to get you excited about," and the reviewer on the Chicago Sun-Times site called it "disconcertingly mediocre."For a show that is not exactly a hit but still pulls in money... | 0.2736 | 0.3708 |
| (Human-written)...You die alone. There is no one with you in the moment when your life fades. But in this, you are not alone. We all share this inherent isolation. We all are born, live, and die alone. There is not one person who ever lived who did not experience this. We all share the empty world the same way... | 0.0774 | 0.0803 |
| (LLM-generated) ... if you learn to trust yourself you can find a special group of friends, a community around you that will help you through the rough days.5. Be a beacon for your friends and family.Remember how amazing it is to have a life mate and family who loves and cares about you? The best thing you can do for someone else ... | 0.1865 | 0.2630 |
| (Human-written)... we look back at the developments so far: 2 August: Samsung unveils its latest flagship model Galaxy Note 7 amid great fanfare in New York. The phone is packed with new features like an iris scanner. The initial response is good and expectations high. It's seen as Samsung's big rival to the upcoming iPhone 7. 19 August... | 0.1189 | 0.1285 |
| (LLM-generated)...a launch on time for the Fourth of July holiday weekend, it may have to rethink the strategy of focusing on India.That's because Indian consumers have always been a difficult group to crack. As with any big, developing nation, Indians are risk takers who aren't particularly eager to buy an Android handset... | 0.1784 | 0.2903 |
| (Human-written)... The manifesto states the Conservatives would introduce a 'funding floor' to protect Welsh relative funding and provide certainty for the Welsh Government to plan for the future, once it has called a referendum on Income Tax powers in the next Parliament. But a Welsh Conservative spokeswoman told BBC Wales: The St David's Day commitment we made to introduce a funding floor for Wales is firm and clear... | 0.0973 | 0.1672 |
| (LLM-generated) ... The Independent has launched its #FinalSay campaign to demand that voters are given a voice on the final Brexit deal. Our petition here The Labour leader, who has said many times he would not sign a hard Brexit, is said to want a more rapid "transition" into post-Brexit British trade arrangements instead. A new spending review could be triggered in July unless a Commons vote is held today to delay it. Downing Street is preparing for the prospect that the UK ... | 0.4478 | 0.4894 |
| (Human-written)... The ground power unit is plugged in. It keeps the electricity running in the plane when it stands at the terminal. The engines are not working, therefore they do not generate the electricity, as they do in flight. The passengers disembark using the airbridge. Mobile stairs can give the ground crew more access to the aircraft's cabin. There is a cleaning service to clean the aircraft after the aircraft lands. Flight catering provides the food and drinks on flight ... | 0.0317 | -0.0172 |
| (LLM-generated)... It also maintains an electrical system, monitors the tire pressure and other factors to ensure that the plane does not run out of fuel. At the airport, the engine is out at the airstairs while the main gear rests on the ground. The helicopter arrives on the runway with the engines running. The pilots pull up to the aircraft and step onto the runway, then shut down the engines and lift the aircraft onto the taxiway. As the air taxis down the runway, a local airport crew member, typically a pilot ... | 0.1331 | 0.1490 |
| (Human-written)... products of the second world made manifest in the materials of the first world (i.e., books, papers, paintings, symphonies, and all the products of the human mind). World Three, he argued, was the product of individual human beings in exactly the same sense that an animal path is the product of individual animals, and that, as such, has an existence and evolution independent of any individual knowing subjects. The influence of World Three, in his view, on the individual human mind (World Two) ... | 0.0481 | 0.1021 |
| (LLM-generated) ... God's existence is a fact of nature; it is a necessary and sufficient condition of everything else. God is to be known and loved; therefore one has a moral right or obligation to believe in God. The secular conception is the conception of naturalism, according to which there is no absolute moral truth or truthlike quality to our beliefs, which is why it is permissible to believe in God. One's belief in God can be replaced by beliefs, however false, which are equally sincere and important, but do not have the intrinsic value ... | 0.1472 | 0.2113 |
| (Human-written)... deep down we all knew this. Just as deep down we knew it was only a matter of time before mans fantasy would got the best of us. It started innocently enough, why not take a few pounds off your avatar, why not skip the traffic to get to work. The changes we made felt so insignificant and made this farce of a life so much more enjoyable. But like with all things it just kept escalating. It wasn't long before the notion of moderation was all but discarded. While knowing that your make believe job had no real meaning why bother going ... | 0.0653 | 0.0597 |
| (LLM-generated)... And the government also is unable to help because we no longer have any antibiotics with which to combat it because we are already too far into the future.The government in the UK is currently trying to use a bit of a law enforcement tactic to get the people to do as they are told, but this is starting to fail, more and more people are starting to realize that they are being lied to, and the government is being revealed as lying to them. We are currently seeing a huge amount of ... | 0.1831 | 0.1983 |
| (Human-written)... The teenager, who was on his way to the conflict in the Middle East, was returned to the custody of his parents while investigations continued, said Mr Dutton. The interception came about a week after two Sydney brothers, aged 16 and 17, were stopped at the same airport on suspicion of attempting to join IS. The brothers, who have not been named, were also returned to their parents ... | 0.1189 | 0.24340 |
| (LLM-generated)... he told ABC radio station AM.\n\nHis comments came after the ABC reported that on December 20, authorities in Singapore and Dubai were investigating the alleged links of several Australians to the terrorism-related activities. That's not to suggest there's been any contact with ISIL in some way in this country or anywhere in the world. Obviously any person associated with the terrorist attack in Paris would have been under scrutiny ... | 0.2144 | 0.2680 |

