# OpenReview forum: "Efficient Detection of LLM-generated Texts with a Bayesian Surrogate Model"
_ICLR.cc/2024/Conference — Submitted to ICLR 2024_

### Official Review · Reviewer_2wLg · 2023-10-26

**Soundness:** 3 good
**Presentation:** 3 good
**Contribution:** 2 fair
**Rating:** 5
**Confidence:** 3

**Summary:**

This paper proposes using a surrogate to replace the true scoring function $p_{\theta}(\mathbf{x})$ for an LLM, due to the high cost of querying the score of a text message $\mathbf{x}$ through the LLM. The authors construct the surrogate function using the Gaussian process and a kernel function that based on the BertScore. The simulations exhibits performance gain for the proposed method over the baseline with the same number of queries.

**Strengths:**

**originality**\
To the best of my knowledge, constructing a surrogate to replace the true score function for LLM to save the expense of LLM queries is novel to me.

**quality**\
The proposed method is simple and straightforward, which is an upside to me. However, there are couple of points that further support the proposed method missing. I will elaborate on those points in the weakness section.

**clarity**\
The paper's presentation is quite clear.

**significance**\
It is significant to propose a method that can reduce the expense of LLM query while maintain high prediction performance for the detection of LLM-generated texts.

**Weaknesses:**

1. The paper establish a scenario that querying LLM is expensive; however, it seems constructing the kernel function with BertScore needs also needs to querying the LLM extensively, and that seems also expensive.
2. One point has not been discussed is the variance of the $ \log p_{\theta} (\tilde{\mathbf{X}})$  (also please use uppercase letter to denote the random variable for the mathematical rigorousness). The proposed method replaces $ \log p_{\theta} (\tilde{\mathbf{X}})$ with a a surrogate function $f$, such that numerous text examples can be used to get the empirical estimation of $E\left[f\left(\tilde{\mathbf{X}}\right)\right]$. On one hand, the variance of such an empirical estimation would be reduced as surrogate function $f$ can be inexpensively accessed; on the other hand, there is bias between $E\left[ \log p_{\theta} (\tilde{\mathbf{X}})\right]$ and $E\left[f\left(\tilde{\mathbf{X}}\right)\right]$. I was hoping to see the analysis of the impact of the variance and bias on the performance of the method, but failed to found them in both methodology and experimental sections.

**Questions:**

Please see the weakness section.

---

> ### Author Response · Authors · 2023-11-17
> **Response to Reviewer 2wLg**
>
> Thank you for your valuable and comprehensive review. Below we address the detailed comments and hope that you may find our response satisfactory and raise your score.
>
> **Question 1: Constructing the kernel function with BertScore needs also needs to querying the LLM extensively, and that seems also expensive.**
>
> We respectfully clarify that the numbers of parameters and FLOPs of BERT-base used for calculating BERTScore are 0.11B and $2.9\*10^{10}$ FLOPs, whereas the GPT-2 for calculating $\log p$ have 1.5B parameters and $3.4\*10^{12}$ FLOPs, and the LLaMA for calculating $\log p$ have 65B parameters and $6\.6*10^{13}$ FLOPs.
>
> Therefore, the query cost of BERTScore is comparatively negligible in the pipeline.
>
> **Question 2:  Hoping to see the analysis of the impact of the variance and bias on the performance of the method.**
>
> Thanks for the constructive comment.
>
> The probabilistic density function $p_\theta$ is **deterministic**. Thus, the mentioned “variance of the $\log p_\theta$” must be w.r.t. the perturbation distribution $q$. However, as detailed in (Mitchell et al., 2023), only $E_{q(x)} [\log p_\theta (x)]$ connects to the probabilistic curvature, the variance $Var_{q(x)}[\log p_\theta (x)]$ is meaningless.
>
> Instead, as suggested by the reviewer, we should care about the variance of **the empirical estimates** of $E_{q(x)} [\log p_\theta (x)]$ and $E_{q(x)} [f(x)]$. For the former, the randomness stems from the data sampling process, and its effect is reflected by the error bars in Figure 2. For the latter, the randomness mainly stems from the samples used to assess $f$ at last, because the typical samples are identified in a nearly deterministic way (except for the initialization). As we use 200 samples to assess $f$, the variance caused by this part is minimal.
>
> To evaluate the bias between $E_{q(x)} [\log p_\theta (x)]$ and $E_{q(x)} [f(x)]$, we can regard our current estimation of $E_{q(x)} [f(x)]$ using 200 samples as accurate and also use 200 samples to estimate $E_{q(x)} [\log p_\theta (x)]$. In our experience, the latter can usually lead to a detection AUROC of more than 98%. In comparison, the detection AUROC of the former with only 10 queries to the source LLM is less than 96%. Thus, the bias is significant. However, it is certain that we can reduce such a bias by incorporating more queries to the source LLMs for $E_{q(x)} [f(x)]$. This is echoed by the results in Section 4.1: once including enough queries, using $E_{q(x)} [f(x)]$ can catch up with the performance of using $E_{q(x)} [\log p_\theta (x)]$.

---

> > ### Comment · Reviewer_2wLg · 2023-11-22
> >
> > Thank you for your reply. Maybe I didn't make my point clearly. For the variance, I meant the variance of $\frac{1}{N}\sum_{i=1}^Nf(\mathbf{X}_i)$,
> >
> > where $f$ is the regression function used to model $\log p_{\theta}$. I do think it is possible to give a quantitative analysis of the variance and the bias of $\frac{1}{N}\sum_{i=1}^Nf(\mathbf{X}_i)$, which can benefit the further understanding of the method.

---

> ### Author Response · Authors · 2023-11-22
> **Further reply**
>
> Thanks for your further clarification. We make the following comments:
>
> 1) **Variance of $\frac{1}{N}\sum_{i=1}^N f(X_i)$.**
>
> It is minimal as we set $N=200$ (i.e., use 200 rephrased texts around the original text to compute an average score). We have performed an empirical study regarding this. Using the setting corresponding to the results in Figure 2a, at the query time of 16 for estimating $f$, we compute the variance of $\frac{1}{N}\sum_{i=1}^N f(X_i)$ over 3 runs for 5 randomly selected texts from the dataset and obtain 0.0006, 0.0003, 0.0011, 0.0013, and 0.0011. In comparison, the mean of $\frac{1}{N}\sum_{i=1}^N f(X_i)$ is -2.2376, -2.5655, -2.4450, -2.2474, and -2.5945 respectively. As demonstrated, the mean is orders of magnitude larger than the variance. We will add more detailed results in the revision.
>
> 2) **Bias of $\frac{1}{N}\sum_{i=1}^N f(X_i)$.**
>
> As $\frac{1}{N}\sum_{i=1}^N f(X_i)$ is an unbiased estimator of $\mathbb{E}_{q(x)} f(x)$, we speculate the mentioned bias corresponds to the gap between
>
> $\frac{1}{N} \sum_{i=1}^N f(X_i) $ and $ \frac{1}{N}\sum_{i=1}^N \log p_\theta (X_i) $.
>
> In our experience, using $200$ samples to estimate the latter usually leads to a detection AUROC of more than 98%. In comparison, the detection AUROC of the former with 10 queries to the source LLM $p_\theta$ for estimating $f$ is less than 96%. Thus, the bias is significant. However, it is certain that we can reduce such a bias by incorporating more queries to the source LLMs for estimating the function $f$. This is echoed by the results in Section 4.1: once including enough queries for estimating $f$, using $\frac{1}{N} \sum_{i=1}^N f(X_i) $ can catch up with the performance of using $\frac{1}{N}\sum_{i=1}^N \log p_\theta (X_i) $.

---

> > ### Author Response · Authors · 2023-11-23
> > **Thanks for reviewing our paper**
> >
> > Dear reviewer 2wLg,
> >
> > As the discussion session draws to a close, we would like to inquire if there are any additional comments or clarifications you would like to make. We are more than willing to provide responses to any inquiries and address any feedback you may have.
> >
> > Thank you for your time and consideration!

---

### Official Review · Reviewer_59P9 · 2023-11-01

**Soundness:** 3 good
**Presentation:** 2 fair
**Contribution:** 3 good
**Rating:** 5
**Confidence:** 4

**Summary:**

This paper aims to address the query inefficiency of methods that detect machine-generated text from Large Language Models (LLMs) by proposing a novel approach that utilizes a Bayesian surrogate model. The proposed approach uses a Gaussian Process model and enhances query efficiency by selecting typical samples based on Bayesian uncertainty and then interpolating scores to other samples.

**Strengths:**

- The proposed approach is quite intuitive and is a relevant step in improving the DetectGPT model proposed previously for detecting LLM generated text.
- The experimental results are conducted on three different datasets as well as in the more realistic setting when there is a mismatch in the source model LLM and the LLM used for detection.
- I appreciate the detailed description of the approach and all the choices made in designing it.

**Weaknesses:**

- The biggest drawback of the paper for me is the *evaluation*. I feel that the current evaluation is lacking in many aspects and feels incomplete:
    - **Limited Open-Source LLMs Considered**: First, only 2 LLMs are utilized, one of which is GPT-2, which is more than 4 years old. The other model used is Llama-1 (65B), although the Llama-2 sets of models have been released for quite some time now. Since there are a number of different open-source LLMs (for obtaining logits in the white-box setting) available with different parameter sizes, the authors should undertake a comprehensive evaluation across many more LLMs: for example, Llama-2, Guanaco, Vicuna, Falcon, MPT, ChatGLM, etc. Even for the experiments of Section 4.3, only smaller models such as GPT-J and GPT-Neo are used. Considering only 2 models (one of which is GPT-2) is not sufficient for evaluation of current performance of the proposed method.
    - **Trends With Respect to Parameter Size**: An obvious question to ask is (irrespective of query size), as the models considered have an increasing number of parameters, does the proposed method become less efficacious? Compared to DetectGPT, how many queries would be required to successfully detect text generated by a more advanced model such as Llama2-70B? I believe the authors could undertake these experiments given that for a number of LLMs (such as Llama) different parameter size models are available (7B, 13B, 70B). It would be interesting to observe performance curves for different classes (with respect to size) of LLMs.
    - **Lack of Analysis on Black-box Models**: I feel that the mismatch setting of Section 4.3 should be further augmented with a black-box setting where state-of-the-art black-box LLMs such as GPT-3.5, GPT-4, PaLM-2, Claude, etc. are analyzed. If the goal of the paper is to truly ensure that LLM generated text is detected, the authors should ideally evaluate on these models via proxy models. As these LLMs are the easiest to use due to a user interface, LLM generated text is most likely to stem from these as sources. It would be beneficial to incorporate some evaluation along these lines.
    - **Overall Lack of Experimental Rigor**: I find some of the claims made in the paper to be hand-wavy and lacking sufficient rigor. For example, on page 7 (Section 4.1 end), the authors state that even under high query budgets, their approach remains effective and comparable to DetectGPT. However, only the WritingPrompts data is used with GPT-2. To clearly make such a point, more experiments should be conducted over all datasets and all LLMs and presented as a table or figure. Then adequate conclusions can be drawn.

- While a minor issue, the paper has many typos and grammatical errors. I believe the authors should go through the paper and correct these in the revision. For example, page 7: "bedget" -> "budget", and page 4: "typicity" means something unrelated to statistics and ML, etc.

**Questions:**

- Why have the authors not considered more LLMs, especially of different parameter size classes in experiments?
- Is there any limiting factor for evaluation of black-box LLMs available only via APIs?
- Please feel free to respond to any of the other weaknesses listed above.

---

> ### Author Response · Authors · 2023-11-17
> **Response to Reviewer 59P9**
>
> Thank you for your constructive comments. Below we address the detailed comments and hope that you may find our response satisfactory and raise your score.
>
> **Question 1: Limited open-source LLMs considered, lack of trends with respect to parameter size and analysis on black-box models.**
>
> Thanks. We first clarify that our existing experiment settings align with DetectGPT (Mitchell et al., 2023). Our current empirical comparisons are fair, having proven our contribution to improving the query efficiency of log-curvature-based LLM-generated text detection. As acknowledged by Reviewer 5xFn, our method “show clear performance improvements over DetectGPT”, and by Reviewer 2wLg, our method ”maintain high prediction performance”. So, we clarify that the lack of the mentioned experiments is not a fundamental limitation of this paper.
>
> Besides, LLaMA-65B is not a trivial LLM, as shown in ([LLaMA 2 paper](https://arxiv.org/pdf/2307.09288.pdf), Table 3), LLaMA-65B can consistently beat LLaMA 2 of 7B, 13B, and 34B across a series of benchmarks (e.g., Commonsense Reasoning, World Knowledge, Reading Comprehension, Math, MMLU, BBH, AGI Eval). As the community of LLM is developing very rapidly, we don’t have enough energy to catch up with all the recent models and *undertake a comprehensive evaluation across many more LLMs: for example, Llama-2, Guanaco, Vicuna, Falcon, MPT, ChatGLM, etc*.
>
> Comparing the results on LLaMA-65B to those on GPT-2, we can see that detecting larger or more advanced models is more challenging for both DetectGPT and our method. This implies that more queries are needed to complete the detection on  LLM with a high parameter size. To further verify these claims, we have started a set of experiments with the series of LLaMA models on the XSum dataset and will offer the corresponding results in subsequent updates (if the running finishes before the end of the rebuttal period) or the final version.
>
> As for black-box detection, our cross-evaluation is sufficient to demonstrate that our method outperforms DetectGPT in black-box scenarios, using the same setting as the DetectGPT paper. We will experiment with the mentioned API-based black-box models in the next version.
>
> **Question 2: Overall Lack of Experimental Rigor**
>
> We totally understand your concern. However, we argue that it is just a case study to prove the convergence trend of our method. The main focus of our method is still the low-budget regime, where our results are from multiple random runs and are statistically rigorous. We will revise the paper to clarify this point.
>
> **Question 3: Many typos and grammatical errors**
>
> Thanks for pointing out this, and we have revised the paper accordingly.

---

> > ### Comment · Reviewer_59P9 · 2023-11-20
> > **Response to Rebuttal**
> >
> > I would like to express my gratitude to the authors for their rebuttal. However, after going through the response, my concerns largely remain. I provide additional details below:
> >
> > * Unfortunately, my concerns regarding limited LLMs and other experiments still remain (parameter size analysis and black-box models). Despite using the same experimental set-up as DetectGPT, I believe more experiments are mandated. If the authors provide additional results before the discussion period, I am happy to take another look.
> > * Llama-2 has been released for quite some time, and as the work aims to generalize results across LLMs, it is important to analyze recent models. A case in point here is that half the experimental evaluation is localized to GPT-2 which is severly outdated. While I understand the authors might face issues with compute, I believe considering more LLMs (and especially recent and more powerful models) is important to judge the efficacy and usefulness of the work.
> >
> > Given the points above, I would like to keep my current score.

---

> ### Author Response · Authors · 2023-11-22
> **Further reply**
>
> Thank you for your patience regarding the results of the LLaMA2 series. Below, you will find the results for XSum:
>
>   |   ||DetectGPT| | |Our method | |
> |---|---|---|---|---|---|---|
> |Query time| 5 | 10 |15 | 5| 10 |15|
> |LLaMA2-7B|0.682|0.719|0.728 |0.704|0.732|0.747|
> |LLaMA2-13B|0.649|0.707 |0.712| 0.664|0.722|0.729|
> |LLaMA2-34B|0.633|0.702|0.708| 0.662|0.715|0.719|
>
> Due to time constraints, we only experimented with up to 15 queries to the source LLM. As shown, both DetectGPT and our method exhibit lower detection AUROC scores as the model size increases. However, our method, using 10 queries, outperforms the DetectGPT baseline using 15 queries. This demonstrates the query efficiency of our method in the LLaMA2 case. It is also worth noting that the LLaMA2 models are indeed more challenging to detect compared to GPT-2.
>
> In the next version, we will investigate if allocating a higher query budget and employing a stronger perturbation model can yield improved AUROC scores. We will also strengthen our cross evaluation of the black-box models. We thank the reviewer again for the constructive feedback!
>
> Best

---

> ### Author Response · Authors · 2023-11-23
> **Thanks for reviewing our paper**
>
> Dear reviewer 59P9,
>
> As the discussion session draws to a close, we would like to inquire if there are any additional comments or clarifications you would like to make. We are more than willing to provide responses to any inquiries and address any feedback you may have.
>
> Thank you for your time and consideration!

---

> > ### Comment · Reviewer_59P9 · 2023-11-23
> >
> > Thank you for the additional results.
> >
> > > This demonstrates the query efficiency of our method in the LLaMA2 case. It is also worth noting that the LLaMA2 models are indeed more challenging to detect compared to GPT-2.
> >
> > While I understand, my concern still holds. It seems that the method does not work well against newer LLMs, which will most likely be used more often (for example, I am not sure if GPT-2 is being used that much any more). This creates a mismatch between the detection approaches and LLMs being used, which I find that the current work is unable to resolve satisfactorily.

---

> > > ### Author Response · Authors · 2023-11-23
> > > **Further clarification**
> > >
> > > We would like to clarify that the reason for the method not performing well against newer LLMs is not due to its inherent limitations. Rather, it is because the number of queries used in our experiments was limited due to time constraints. Besides, the paper focuses on demonstrating the superiority of our method over DetectGPT in the low-budget regime. However, it is worth noting that we can improve the AUROC by using a larger number of queries, as demonstrated in the DetectGPT paper. In the next version, we will include experiments with a higher query count to address this aspect.

---

### Official Review · Reviewer_5xFn · 2023-11-01

**Soundness:** 3 good
**Presentation:** 3 good
**Contribution:** 3 good
**Rating:** 6
**Confidence:** 3

**Summary:**

The authors propose a new approach to use a Gaussian Process (GP) surrogate model to learn the sample distribution of LLM output to effectively sample pertrubations to detect text generated by LLM. THey show that their approach outperforms detectGPT approach in in number of queries needed to effectively detect text. They also show impovements in AUROC.

**Strengths:**

Simple yet effective approach to identify data generated through LLM.
Show clear performance improvements over DetectGPT.

**Weaknesses:**

Experiments section lacks other baselines.
Minor: Some Figure label text in experiments can be improved.

**Questions:**

While the experiments are good, Why not have results similar to Detect GPT? The numbers in their paper and here dont match up.
What is the performance like if you used some other model that is not GP as surrogate?
Did you look result if you used a encoder model as classifier and trained a bit as its easy to generate the data for this? (May be a bit out of scope. just curious)

---

> ### Author Response · Authors · 2023-11-17
> **Response to Reviewer 5xFn**
>
> We appreciate the reviewer for the positive feedback and the acknowledgment that our method is simple and effective. Below we address the detailed concerns.
>
> **Question 1: Regarding other baselines**
>
> Thanks for the comment. As stated in the first paragraph of Section 4, we mainly compare our method to DetectGPT (Mitchell et al., 2023) because (i) both works adopt the same detection measure (probability curvature), and (ii) DetectGPT has proven to defeat prior zero-shot and supervised methods consistently. We recently noted some concurrent works [1, 2] and will add discussions and/or empirical comparisons with them in the final version.
>
> **References:**
>
> [1]  Guo, Biyang, et al. "How close is chatgpt to human experts? comparison corpus, evaluation, and detection." arXiv preprint arXiv:2301.07597 (2023).
>
> [2]  Bao et al. “Fastdetectgpt: Efficient zero-shot detection of machinegenerated text via conditional probability curvature.” arXiv preprint arXiv:2310.05130(2023)
>
>
> **Question 2: Why not have results similar to Detect GPT? The numbers in their paper and here dont match up.**
>
> Thanks. As our main contribution is to improve the query efficiency of probability curvature-based detectors like DetectGPT, *we are primarily concerned with detecting under a low query budget* (stated in the first paragraph of Section 4). We also *admit that our method would perform similarly to DetectGPT if queries to the source model can be numerous*.
>
> Regarding the numbers, those in the paper of DetectGPT correspond to a query number of 1000, while those in our paper correspond to 1-15 query times.
>
> The reported numbers prove that the effectiveness of DetectGPT is significantly reduced at a low query budget, and our method addresses this issue.
>
> **Question 3: What is the performance like if you used some other model that is not GP as surrogate?**
>
> Thanks for the suggestion. However, we would like to clarify that the specification of the mentioned *other model* is non-trivial. As stated in Sec 3.3, *the surrogate model f is expected to be trained in the low-data regime while being expressive enough to handle non-trivial local curvature and not prone to overfitting. Additionally, the model should inherently incorporate mechanisms for typical sample selection*. Thus, we choose the GP and speculate that parametric models like NNs may not be applicable here. We will continue to explore other choices, such as neural processes and implicit processes, in future work.
>
>
> **Question 4: Did you look result if you used a encoder model as classifier and trained a bit as its easy to generate the data for this?**
>
> Thanks for the question. We would like to clarify some key points: 1) The DetectGPT paper has already demonstrated that the original DetectGPT outperforms the RoBERTa encoder-based classifier. Given our superior performance compared to DetectGPT, it can be inferred that our model would also outperform RoBERTa at higher query numbers. 2) Supervised detectors are often susceptible to biases from training data and entail higher training costs. In contrast, our method can better generalize and is training-free.

---

> > ### Comment · Reviewer_5xFn · 2023-11-23
> > **Thank you for your response**
> >
> > THank you for the clarifications. It will be good to include some of these details in the paper. Thanks.

---

### Meta-Review · Area_Chair_3riW · 2023-12-07

**Metareview:**

The paper aims to enhance the efficiency of detecting text generated by large language models (LLMs) using a log-curvature-based approach. Reviewers raised concerns primarily around the paper's experimental rigor and the range of models tested. While the paper aligns with existing methods like DetectGPT, it is criticized for limited experimentation with only a few open-source LLMs, notably GPT-2 and LLaMA-65B. Reviewers suggest expanding the range of LLMs tested, including more recent and diverse models, to provide a comprehensive evaluation. Additionally, there's a call for more robust analysis in both black-box and white-box settings and a deeper exploration of the method's efficacy across different LLM parameter sizes. The authors have responded by emphasizing their alignment with DetectGPT and promising further experiments and updates in the final version.

**Justification For Why Not Higher Score:**

A higher score isn't justified due to the paper's limited scope in experimental evaluation. The primary concern is the narrow range of LLMs tested, particularly the reliance on older models like GPT-2 and not including more recent and varied LLMs. This limited range hampers the paper's ability to demonstrate the effectiveness of its proposed method across a wider spectrum of LLMs, especially more advanced models that are more relevant in current research and applications. Furthermore, the lack of rigorous analysis in different settings (e.g., black-box models) and insufficient exploration of trends relative to LLM parameter sizes leaves critical questions unanswered, hindering the paper's contribution to the field's understanding of LLM-generated text detection.

**Justification For Why Not Lower Score:**

NA

---

### Decision · Program_Chairs · 2024-01-16

Reject